# Tensile Strength Essay Comparing Three Different Platelet-Rich Fibrin Membranes (L-PRF, A-PRF, and A-PRF+): A Mechanical and Structural In Vitro Evaluation

**DOI:** 10.3390/polym14071392

**Published:** 2022-03-29

**Authors:** Mara Simões-Pedro, Pedro Maria B. P. S. Tróia, Nuno Bernardo Malta dos Santos, António M. G. Completo, Rogerio Moraes Castilho, Gustavo Vicentis de Oliveira Fernandes

**Affiliations:** 1Faculty of Dental Medicine, Universidade Católica Portuguesa, 3515-320 Viseu, Portugal; marasimoesp@gmail.com (M.S.-P.); pmbpst97.pt@gmail.com (P.M.B.P.S.T.); berna74@gmail.com (N.B.M.d.S.); 2Centre for Mechanical Technology and Automation, TEMA—University of Aveiro, 3810-549 Aveiro, Portugal; completo@ua.pt; 3Department of Periodontics and Oral Medicine, University of Michigan, Ann Arbor, MI 48104, USA; rcastilh@umich.edu

**Keywords:** platelet-rich fibrin, tensile strength, resistance, rupture, fibrin, microscopic analysis, centrifugation

## Abstract

Predictable outcomes intended by the application of PRF (platelet-rich fibrin) derivative membranes have created a lack of consideration for their consistency and functional integrity. This study aimed to compare the mechanical properties through tensile strength and analyze the structural organization among the membranes produced by L-PRF (leukocyte platelet-rich fibrin), A-PRF (advanced platelet-rich fibrin), and A-PRF+ (advanced platelet-rich fibrin plus) (original protocols) that varied in centrifugation speed and time. L-PRF (n = 12), A-PRF (n = 19), and A-PRF+ (n = 13) membranes were submitted to a traction test, evaluating the maximum and average traction. For maximum traction, 0.0020, 0.0022, and 0.0010 N·mm^−2^ were obtained for A-PRF, A-PRF+, and L-PRF, respectively; regarding the average resistance to traction, 0.0012, 0.0015, and 0.006 N·mm^−2^ were obtained, respectively (A-PRF+ > A-PRF > L-PRF). For all groups studied, significant results were found. In the surface morphology observations through SEM, the L-PRF matrix showed a highly compact surface with thick fibers present within interfibrous areas with the apparent destruction of red blood cells and leukocytes. The A-PRF protocol showed a dense matrix composed of thin and elongated fibers that seemed to follow a preferential and orientated direction in which the platelets were well-adhered. Porosity was also evident with a large diameter of the interfibrous spaces whereas A-PRF+ was the most porous platelet concentrate with the greatest fiber abundance and cell preservation. Thus, this study concluded that A-PRF+ produced membranes with significant and higher maximum traction results, indicating a better viscoelastic strength when stretched by two opposing forces.

## 1. Introduction

Platelet concentrates have been widely utilized in medicine and dentistry owing to their rapid angiogenic stimulation ability and potential for tissue regeneration [1]. Platelet-rich fibrin (PRF) was proposed as a second-generation platelet concentrate, seeking to minimize contamination risk by avoiding the added use of anticoagulants or animal-derived thrombin. It was first described by Choukroun and collaborators (2000) and named as leukocyte platelet-rich fibrin (L-PRF) [2]. Briefly, its production protocol comprises the collection of autologous patient blood and immediate centrifugation to trigger the activation of the platelets and fibrin polymerization. Centrifugation forms a complex three-dimensional scaffold where the main broad cellular spectrum components are centered, including platelets, growth factors (VEGF, PDGF, TGF-β1, EGF, IGF-I, and HGF) (Appendix A), leukocytes, and cytokines, which are known to play a fundamental role in the interaction of regeneration and wound healing [3,4].

PRF is a matrix that supports angiogenesis, proliferation, cell differentiation, and chemotaxis within the fibrin, acting as a biomimetic reservoir for cells and cell signaling [5]. Therefore, it is appliable in periodontal regeneration (root coverage of gingival recession), alveolar ridge preservation, sinus lifts, ulcers/skin necrosis, chronic wounds, plastic and reconstructive surgery, musculoskeletal lesions, and tendon injuries [3,4,6,7].

### 1.1. Low-Speed Centrifugation Concept (LSCC)

Leukocytes and platelets require a reduction of rotational centrifugal force (RCF) in order to preserve their source of growth factors (GFs). Recent evidence has demonstrated that the consequences of a high RCF not only decreases the number of cells but also negatively influences the ability to release GFs [8,9].

Platelets recover from the centrifugation speed during the resting time. However, mainly larger and metabolically active ones with prothrombotic potential are removed in high-speed centrifugation, which is shown by a light transmission aggregometry (LTA) analysis, possibly leading to a decrease in aggregation [10]. TGF-β is an exception that g-forces [11] do not influence.

Blood coagulation is a rapid process that initiates even before the tubes are exposed to centrifugation. Plastic tubes contain siliceous substances that favor this clotting activation so the recommended time between the blood drawing and the start of centrifugation is 60–90 s per 5 collecting tubes. Otherwise, the size of the PRF membrane is affected, suffering a significant reduction [12]. TGF-β is simultaneously released when the platelets are activated and bond to the newly formed fibrin-rich extracellular matrix, which is protected from centrifugal forces [11].

### 1.2. Advanced Platelet-Rich Fibrin (A-PRF)

Over the past few years, modifications to the original L-PRF protocol such as updating the centrifugation time and g-force have led to the emergence of A-PRF, also developed by Choukroun, which introduces a decrease of the rotation speed whilst the centrifugation time is increased [13]. This achievement demonstrates improved mechanical and biological properties regarding L-PRF and better outcomes. The use of low g-forces for A-PRF has revealed a different pattern of cell distribution in immunohistochemical studies. A greater number of platelets and leukocytes are widely dispersed throughout the clot. This finding also correlates with the interesting increase of granulocytic neutrophil populations in the distal part of the collected supernatant that are not detected following high centrifugation forces [14,15].

Based on the current published scientific papers, the reticular and porous microstructure of A-PRF explains its elasticity. In turn, these inherent factors enable the idea of an encapsulation capacity for more cellular components in the interfibrillar spaces of the membrane, which has already been proven [16]. An increased migration as well as the proliferation of fibroblasts and mRNA collagen levels have also been reported [17]. 

### 1.3. Advanced Platelet-Rich Fibrin+ (A-PRF+)

Another new variant known as A-PRF+ was suggested by Fuijoka-Kobayashi et al. [14] who reduced the rotation time whilst maintaining the same force used in the A-PRF protocol. This is thought to have distinct advantages despite the slight decrease in time, positively impacting on the preservation of cells in the formed clot and, consequently, a further improvement in the characteristics of A-PRF+ compared with those mentioned above from A-PRF.

The highest percentage of cells that can be collected has been already proven as well as the higher uniformity of platelet distribution and porosity than described in A-PRF. This happens in such a way that it is possible to establish a balance between the trapped cells and chemotaxis, migration, proliferation, and degradability closely linked to the sustained release of growth factors [18].

The authors validated the hypothesis that changes in the mechanical properties of A-PRF and A-PRF+ may indeed induce differences in the release of certain groups of growth factors. For example, significantly higher values of PDGF and TGF-β1 were released within the scaffold on day 7 and 10 of a limited period of measurement studies; for EGF, the maximum release range in both membranes happened only at the early time point of 24 h and was more marked in A-PRF+ and VEGF appeared to be one of the most accumulated essential growth factors in A-PRF+, possibly explained by the affinity of VEGF with the amounts of fibrin and fibrinogen in the organized mesh [1,5,16].

The considered LSCC used in A-PRF and A-PRF+ creates pores with a higher diameter in the fibrin network, allowing cells—together with a few vessels—to perfuse the peripheral edges of the scaffold [19].

Mechanical interest has increased in response to the unbalanced gap in the need for matrices with consistency and functional integrity, allowing for proper handling, suturing without breaking, and the enhancement of biological space for the continued release of cells and GFs over time [8,20]. Even though blood composition is specific and individual, PRF-based matrices are reproducible systems with a distribution independent of the characteristics of the donor, making it possible to perform a clot analysis under appropriate conditions for study [21]. Therefore, the goal of this article was to assess and compare the variability of the resistance with the tension of three different types of PRF membranes, which were produced through the original L-PRF, A-PRF, and A-PRF+ protocols. The positive hypothesis of this study was to ascertain whether the use of the lower-speed centrifugation concept (LSCC) and a reduced time for preparation provided a greater tensile property to suggest that this blood concentrate was more able to withstand stress when applied in surgeries.

## 2. Materials and Methods

### 2.1. Blood Collection and Membrane Preparation

This study followed the Helsinki Declaration of 1975 as revised in 2013 and the study was approved by the Ethics Committee of the Universidade Católica Portuguesa, Viseu, Portugal (number 522020, 26 March 2020). Peripheral blood was collected with a butterfly needle from a single healthy donor who was male and 23 years old with no history of a use of anticoagulants or immunosuppressors, thus avoiding the bias of analysis. These trials were adequately spaced between 20 days for the recovery and comfort of the donor with 9 membranes produced on the experimental day, totaling 5 days for scientific development (D1 to D5).

A sample size of 44 PRF membranes were tested in the current study. Therefore, to detect a minimum significant difference from the tensile test of the membranes, it was necessary to use at least 12 membranes/group based on a previous study of our group (Pascoal et al., 2021), which found means between the groups of 0.01923 and 0.02885 from L-PRF and A-PRF, respectively, using a two-tailed test of variance, α = 0.05, power of 95%, and a standard deviation of 0.006760.

For each protocol, membranes of L-PRF (n = 12), A-PRF (n = 19), and A-PRF+ (n = 13) using specific tubes (9 mL sterile glass-coated plastic tubes, BD Vacutainer^®^, Plymouth, PL6 7BP, UK), were obtained, giving rise to a variation in the number of membranes collected due to a loss of structure in the protocol errors prior to the traction test. The tubes were placed in the centrifuge in pairs to ensure an adequate balance. When pairs were not obtained, an extra tube was used with water to equilibrate the system. The total period from harvest to the beginning of the centrifugation did not exceed 2 min, attempting to preserve the polymerization and size outcomes [22]. The centrifuge was the same for all protocols, an IntraSpin™ centrifugation device (Intra-Lock, Boca Raton, FL, USA) (33° rotor angulation, 50 mm radius at the middle of the tube, 80 mm at the maximum with 40 mm maximum and 40 mm at the minimum).

At the environmental temperature of 20 ± 2 °C, the L-PRF membranes were prepared at 2700 revolutions per minute (rpm), 408 g, for 12 min (Khorshidi et al., 2018); for the A-PRF membrane, either the centrifugation time or speed was different, following the original values of 1500 rpm (126 g) for 14 min [13,23]. The preparation of A-PRF+ was different from A-PRF for the centrifugation time; this was 1500 rpm (126 g) for 8 min. After completing this step, the lid was opened, exposing the tubes to room temperature, and they were left to rest for 15 min to improve the thickness of the filaments [24].

After the careful removal of the red blood clots, the fibrin portion associated with the buffy coat attached to the PRF was then separated (Figure 1A). Consequently, the PRF was gently compressed by gravity for 5 min and slightly pressured until the metal cover was completely closed, following the manufacturer’s recommendation. 

### 2.2. Scanning Electron Microscopy (SEM)

For the SEM analysis, six additional membranes of A-PRF and A-PRF+ were utilized. Three membranes of L-PRF were also added as a control group to evaluate any distinct high and low centrifugation results.

After the preparation, the PRF clots were sectioned with a scalpel into fragments of an equal length (5 × 5 mm) to have representative samples of the three layers of each membrane. The sections for the microscopic evaluation were then distributed, starting with the removal of the lower fragment adjacent to the fraction of red blood cells (RBCs) encompassing the buffy coat; the second was from the middle segment of the membrane and the third was from the lower-upper portion (Figure 2).

Each sample was fixed with 2.5% neutralized glutaraldehyde (2 h, 4 °C) (Figure 3A), postfixed with a 0.2 M sodium cacodylate buffer solution and 1% osmium tetroxide (2 h, 4 °C), and finally dehydrated in a series of ethanol solutions (ranging from 70 to 100%) and hexamethyldisilane (Figure 3B). The materials were metalized with silver (Figure 3C) and observed at a voltage acceleration of 15 kV using a scanning electron microscope (Hitachi SEM S-4100, Hisco Europe, Ratingen, Germany) as previously described [25].

### 2.3. Traction Assay

A glass mold was specially designed and manufactured to make the fibrin specimens identical in size, permitting the membranes to be cut with the same dimensions (10 × 15 mm) prior to the traction test (Figure 1B). Clamps gripped the samples at each end such that the initial apparent gauge length (the distance between clamps) was set to 13 mm for all the samples tested. A traction test was performed with Shimadzu MMT-101N equipment (Shimadzu Corporation, Japan) (Figure 1C). The maximum tensile strength and tensile strain at the break applied divergent vertical forces and always used the same movements and force. The maximum value for traction using this equipment was set to 12 mm or up to the rupture [26]. The mechanical assay was performed at the Centre for Mechanical Technology and Automation (TEMA) at the University of Aveiro (Portugal).

### 2.4. Data and Statistical Analysis

The mechanical properties were plotted by a force applied to the membranes per section area (N·mm^−2^) and the tensile of the membrane at a stretching speed of 1 mm/minute, obtaining a stress–strain curve recorded until the rupture or until the limit of the maximum traction of the machine was reached. This provided us with the maximum elasticity and tensile strength. Data were collected using Microsoft Excel (v. 16.50, Microsoft, Redmond, WA, USA) and GraphPad Prism (v. 8.2.1, GraphPad software, Inc., San Diego, CA, USA).

All statistical analyses were performed using GraphPad Prism software. The values are presented as a mean ± SD in the legends of the figures. All data were firstly analyzed through a Kolmogorov–Smirnov test (normality). A Kruskal-Wallis test was then developed with all groups having a significant statistical level if *p* ≤ 0.05.

## 3. Results

### 3.1. Data Analysis

This study proposed to compare three platelet concentrates (preparations), L-PRF, A-PRF, and A-PRF+, to find possible differences in the mechanical properties. It was intended to primarily evaluate the traction of each membrane characterized by two parameters: (a) the maximum stress on the stress–strain curve; and (b) the stress at rupture. As shown in Figure 4A, it was essential to determine the maximum strength of the membranes that could, in a few cases, double their length from the original cut size when subjected to traction. We also generated a statistical representation of the average resistance, as shown in Figure 4B. 

The tensile test evaluation spotted all membranes of each concentrate, indicating that there was a statistically significant difference for the maximum traction as the average traction between the protocols A-PRF+ and L-PRF (*p* < 0.001) were similar to the occurrence between A-PRF and L-PRF (*p* < 0.001) as well as between A-PRF and A-PRF+ (*p* < 0.01) (Figure 4C).

In reference to the maximum traction (Figure 4A), L-PRF obtained a value of 0.0010 N·mm^−2^; A-PRF was 0.0020 N·mm^−2^ and A-PRF+ was 0.0022 N·mm^−2^. When the average resistance to traction was recorded, A-PRF obtained 0.0012 N·mm^−2^ and A-PRF+ obtained 0.0015 N·mm^−2^ whereas the L-PRF group obtained 0.006 N·mm^−2^. This demonstrated a superior resistance for A-PRF+ followed by A-PRF and L-PRF (A-PRF+ > A-PRF > L-PRF) (Figure 4C).

### 3.2. Morphology Characterization

Figure 5 shows several representative images from the surface microstructure of the fibrin clots that originated from each protocol by scanning electron microscopy (SEM). Based on the SEM analysis, the L-PRF matrix (2400 rpm, 12 min) showed a highly compact surface with thick fibers present within interfibrous areas as well as limited and reduced space for microvascularization (Figure 5A,B). In addition, the severe destruction of red blood cells and leukocytes was visible (Figure 5C). 

The A-PRF protocol (1500 rpm, 14 min) showed a dense matrix (Figure 5D) composed of thin and elongated fibers that followed a preferential and orientated direction (Figure 5E,F) in which the platelets were well-adhered. Porosity was also evident with a large diameter of the interfibrous spaces (Figure 5E). Traces of silica microparticles were detected lying on the surface, as shown in Figure 5D (yellow arrow), provided from the tube of processing.

For A-PRF+ (1500 rpm, 8 min), the number of cross-linking fibers was even more noticeable (Figure 5G–I) as the presence of intact lymphocytes adhered to the surface of the mesh (Figure 5G), which were entrapped together with clusters of platelets (Figure 6A). It was possible to observe platelets adhered to the fibers (Figure 6C) and a lower-upper portion (#1) with more compact fibers. However, the middle and lower parts (#2 and #3) presented matrices with a greater porosity and irregularity of the surface with exophytic portions of the fibers as well as a low destruction of the cellular findings.

## 4. Discussion

To the best of our knowledge, this pioneering study reports the traction range of the membranes originated by three different PRF derivates (L-PRF, A-PRF, and A-PRF+), the second generation of platelet concentrates. The results of this study support the notion that there are clinically relevant differences in the mechanical properties among the produced membranes. The structural integrity of the biomaterials is a factor that suggests its contribution to the prolonged release of several growth factors. It should provide a non-immunogenic microenvironment that, through the properties of the network, enhance migration and cell adhesion [16]. Recent research developed by Lourenço et al. [27] confirmed that the microenvironment created by the PRF membrane architecture is directly related to its bioactivity whilst degradation takes place [28]. The resilience of the dense fibrin of the matrices in the final preparation allows for a long and slow release of multiple molecules [29].

The respective fibrin meshes produced by each protocol were placed in an angled centrifuge (IntraSpin^®^ centrifuge). Pascoal et al. [30] referred to the higher traction ability of A-PRF membranes concerning L-PRF. Thereby, this study aimed to verify and validate these results regarding A-PRF and L-PRF, and they were in complete agreement with Pascoal et al., introducing the reliable evaluation of the mechanical effects in the most recent protocol, A-PRF+.

### 4.1. Tensile Assay

In the tensile test, the overall evaluation tended toward discovering a quantitative order in the resistance parameter, demonstrating the maximum tensile strength results for A-PRF+ followed by A-PRF. Although Ravi and Santhanakrishnan [5] found extremely high values for A-PRF (362.565 ± 5.15 MPa), these values did not coincide with those of this study, justifying different protocol settings. The samples were produced with a tabletop centrifuge (Remi R4C) and cut according to a higher length mold to perform the tensile test using a universal testing machine. 

However, there was compliance with the scenario of a notable increase in the elastic capacity when LSCC was implemented [5]. In the present work, A-PRF obtained an average of 0.0020 MPa and A-PRF+ obtained an average of 0.0022 MPa, depending on a much higher tensile force to suffer deformation or rupture only by the reduction of centrifugation speed and time. Although both scored less than the obtained 0.2 ± 0.06 MPa resistance of L-PRF prepared by Khorshidi et al. [28], the results carried out by the methodologies adopted by Pascoal et al. [30] recorded an A-PRF value of 0.0752 MPa for maximum traction, which was higher than in this study but still surpassed the reach of the membranes produced by L-PRF. The motive for this result to be more elevated was the cut size of the membrane, which was more significant than the standardized measures in our research.

The effect felt in the A-PRF+ group was approximately similar to A-PRF. However, they cannot be entirely compared with the previously reported studies because this study, as far as it is shown in the literature, was the first to elucidate the inherent mechanical properties of this derivative using strictly standard protocol techniques as originally described in [14,15].

Lam et al. [31] refer to the action of platelets on a clot and detected a positive impact which, in addition to fibrin, was a source of reinforcement of the adhesive properties. This fact suggests that the significant increase in elasticity and tensile strength found in the low RCF protocols performed may also result from a more dispersed platelet distribution along with the clot. In agreement with a previous report [32], which concluded that a slight decrease in the centrifugation time largely preserved the cellular contents, a higher amount and an even more homogeneous distribution of platelets was presented [16,32]. Therefore, this suggests that A-PRF+ could be preferable as a scaffold for tissue repair when a high tensile strength is an issue such as in wound closures, the distribution of mechanical forces, and the maintenance of the suture.

### 4.2. SEM Analysis

Through a SEM analysis, the morphology of the PRF membrane surface of each protocol was evaluated to understand the characteristics of the fibrin fibers, which are essential for mechanical and biological clot homeostasis. Thus, a large fibrin mesh in A-PRF+ was found that was arranged in organized layers that, when magnified, suggested a superior resistance to traction. 

A remarkable similarity was observed through a microscopic analysis between A-PRF and A-PRF+ regarding porosity formed by porous tangled spaced fibers with a notoriously high density of connections directed longitudinally and laterally. The anatomy of the fibers in the low centrifugation protocols commonly revealed a thin biotype although A-PRF+ appeared to have a higher polymerization maturity despite its thin thickness.

L-PRF presented a matrix of dense and strongly polymerized fibrin but with less porous and thicker fibers in contrast to its counterparts. According to Li et al. [33], the thickness of fibers suggests a weaker connection between the protofibrils, tending toward a lesser modulus of elasticity. Experimental evidence corroborates the lower value of the mechanical properties with the degradation process observed in L-PRF [5]. It can be assumed that a coagulum composed of many thin fibers is more auspicious to undergo slow lysis by the fibrinolytic system, leading to the belief and suggestion that A-PRF+ might have a longer dissolution rate than L-PRF, lasting approximately the same amount of time as A-PRF according to its similar characteristics.

The key factor that provides membrane resistance is its structural integrity, which allows it to have a long sustaining mechanism in order to improve clinical outcomes [16,26]. The findings of this study showed an increased porosity and strong fibrin architecture for A-PRF+, which could also help to explain the possible prolonged release of growth factors as reported in other studies [17,19]. The cell damage and destruction could be allocated to the vibrations inherent in the centrifugation step [10,33]. Of the three derivatives, A-PRF+ was the one that showed the least amount of damage without the destruction of the content. Although a greater number of lymphocytes were observed in L-PRF, signs of their altered or damaged form were present, similar to what was found in A-PRF. Further analyses are required to assure these non-representative findings.

The findings of this study confirmed the hypothesis that A-PRF+ is more resistant than A-PRF with a statistically significant result; it had a more porous microstructure in which its fibers had small physiognomic differences from those revealed in A-PRF. 

In the literature, the only article that opposed these findings was the experiment carried out by Dohan Ehrenfest et al. [34], which noticed a disorganized fibrin network with thin and weak fibers and greater cell destruction in A-PRF, considering that it has a lower biological signature. Nonetheless, the empirical results reported herein must be seen, considering the limitations that should be addressed in future research.

### 4.3. Silica-Coated Tubes

In the design of the current study, the first concern was the use of plastic vacuum blood collection tubes coated with silica activators on the inner walls. Without anticoagulants, the fibrin clot formation was immediately triggered by contact with the silica particles. The possible risk of biological contamination has been associated with concentrates collected in glass or plastic tubes with this activator [35]. A small fraction of microparticles can inevitably detach from the tube walls during centrifugation and be suspended or retained in the lower layer of the fibrin. This may have consequences for the host organism that are not yet fully known.

Even in residual amounts, these microparticles are released as the membrane degrades, raising questions about its possible help to develop cytotoxic or inflammatory effects [35,36]. Glass tubes or titanium tubes to produce titanium-prepared platelet-rich fibrin (T-PRF) may be a more biocompatible alternative and, so far, there are indications that they can form a clot that is clinically similar to one formed in glass or plastic tubes [35]. However, it is necessary to investigate whether there is maintenance or improvement in the reported properties.

### 4.4. Final Considerations

Other analyses validated the importance of the results presented in this study of tensile strength and stability created by the fibrin network [5,25,29,37]. The recognition of the properties that seem to be enhanced by the reduction of centrifugation—more precisely, the increased resistance at rupture, the amount of microporosity, and the flexibility of the membrane—has aroused a growing interest in several areas of medicine as a promising treatment in complex and delicate defects to repair.

In medicine, treatment possibilities in which there are a lack of adequate biomechanic repairs are tympanic membrane perforations, the regeneration of focal articular cartilage, tendon remodeling, and the repair of muscle, bone defects, or other soft tissue injuries [26,27,38]. In dentistry, the application of these concentrates has shown favorable postoperative conditions to make their use routine. PRF was a more beneficial and less invasive alternative in a meta-analysis study involving periodontal surgery than a connective tissue graft, which is considered to be the “gold standard” for the root coverage of gingival defects that reduces discomfort for the patient [38]. It has also provided clinical results comparable with the use of bone grafts with a gain in the attachment, resolution, and regeneration in two to three walls of infra-bone defects, proving to be biologically safer and practical as well as a less expensive alternative [39]. In oral and maxillofacial surgery, the effects are already strongly documented [40] and include the reduction of swelling and pain in third molar extractions, the preservation of the alveolar ridge by decreasing the bone resorption, a reduction in healing time in sinus lift procedures that favor optimal bone healing, and the acceleration of osseointegration in dental implants. 

An advanced experiment joins the reviews above, managed by Xin and collaborators [41], which was able to display a total repairment of the perforated Schneiderian membrane and a greater amount of new bone formed under the area through the application of A-PRF in a sinus lift model.

It was noticeable that A-PRF+ gave rise to a more resistant environment. The decrease in time and the pull-down effect created by the centrifugation forces increased the total number of cells left and determined the degree of bonds between the molecules [14,16]. This resistance can be considered to be the main factor that influences the performance of the membrane as a whole; in directing the differentiation of stem cells that are mechanosensitive to the surrounding environment, the release of growth factors or other adhered nanoparticles with a pharmacokinetic potential increase. A delayed speed of degradation occurs because mechanical deformation occurs in layers and is proportional to the rates of nutrient consumption by the cells [18,20,42].

One of the study limitations was the lack of a morphometric analysis. Thus, the SEM images could have a subjective evaluation. Further studies should be conducted to morphometrically evaluate the microscopic images as other parameters, as in the case of tube-rotor angulation using horizontal centrifugation, which is less traumatic for the cells and has a better capacity for separation by densities [43]. The effects that it may have on PRF resistance are not yet known. 

Future clinical studies applying this biomaterial (A-PRF+) could be developed such as for root coverage treatment in periodontics (dentistry). As A-PRF+ has not been used before in this type of procedure, and observing the recent result found by Fernandes et al. [44] who considered autologous platelet concentrates of the second generation a feasible substitute for the gold standard (connective tissue), the results of the current study may improve the knowledge in this field.

## 5. Conclusions

Despite a few limitations, it was possible to affirm from the mechanical parameters that were tested a significantly greater tensile strength in the membranes produced with the A-PRF+ protocol, making this type of membrane the most favorable to be sutured and handled. Moreover, A-PRF+ showed an interesting surface morphology that balanced the proportion of porosity with the arrangement and number of fibers that may contribute more to membrane resilience than L-PRF and A-PRF. However, more accurate analyses are required to evaluate the direct impact of this scaffold structure type.

Future in vivo studies are suggested to evaluate the maintenance of these properties and how they are may have an advantageous interference on tissue regeneration in short- and long-term analyses, combining the suggested methodological considerations.

## Figures and Tables

**Figure 1 polymers-14-01392-f001:**
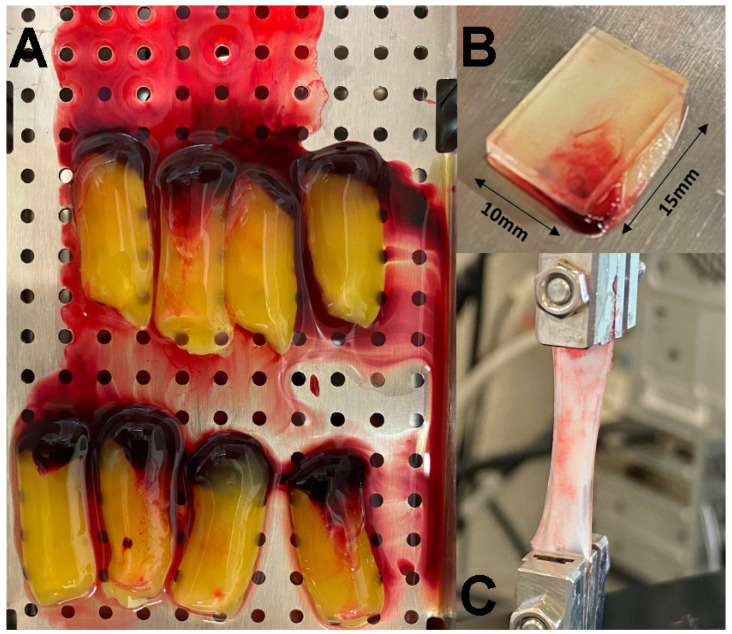
(**A**) Clots were collected from the centrifuged tubes and placed in the Xpression box kit. (**B**) A glass mold was designed and manufactured to make the fibrin specimens identical in size and dimensions (10 × 15 mm^2^). (**C**) Clamps gripped a sample for the traction assay.

**Figure 2 polymers-14-01392-f002:**
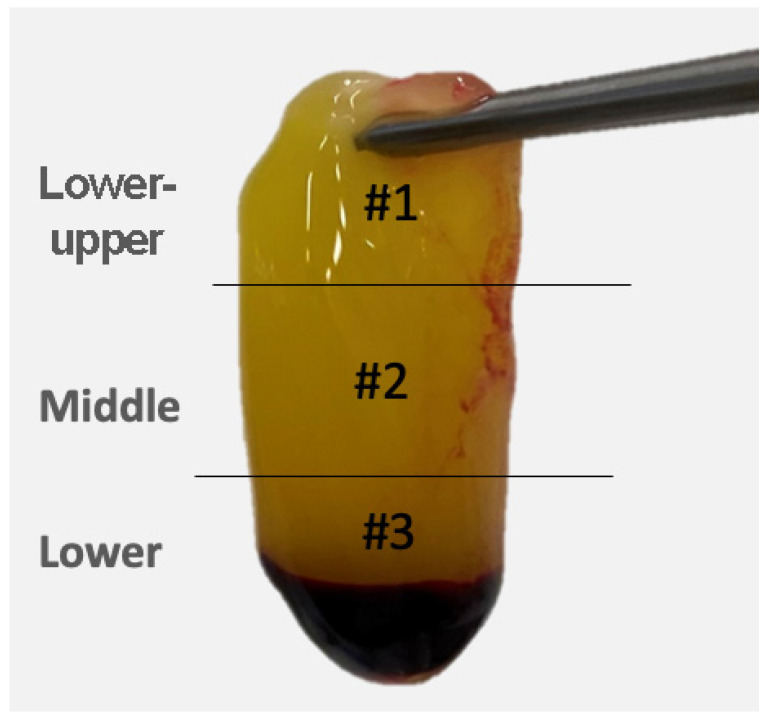
PRF membrane. #**1**. Lower-upper portion. #**2**. Middle portion. #**3**. Lower portion.

**Figure 3 polymers-14-01392-f003:**
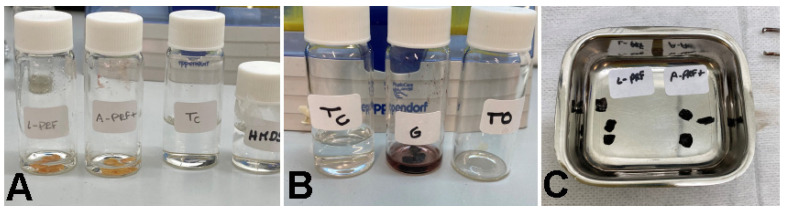
SEM preparation. (**A**) Each sample was fixed with 2.5% neutralized glutaraldehyde. (**B**) Postfixed with 0.2 M sodium cacodylate buffer solution and 1% osmium tetroxide and dehydrated in a series of ethanol solutions (ranging from 70 to 100%) and hexamethyldisilane. (**C**) The materials were metalized with silver.

**Figure 4 polymers-14-01392-f004:**
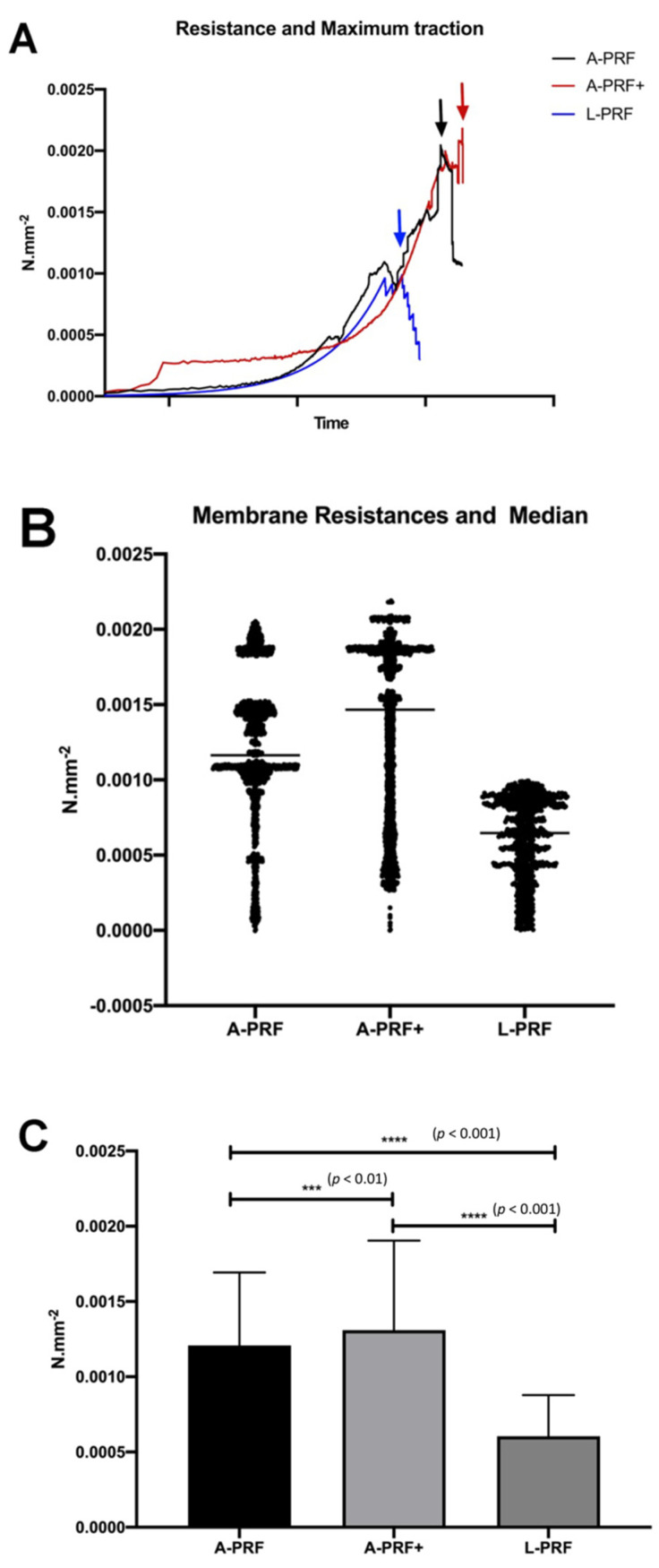
(**A**) Resistance and maximum traction where A-PRF+ reaches a higher strength value compared with A-PRF and L-PRF tested. (**B**) Average and standard deviation (SD) comparing all groups analyzed. (**C**) Membrane resistance and median show the variability of results felt mainly in the L-PRF group and more consistently in A-PRF+.

**Figure 5 polymers-14-01392-f005:**
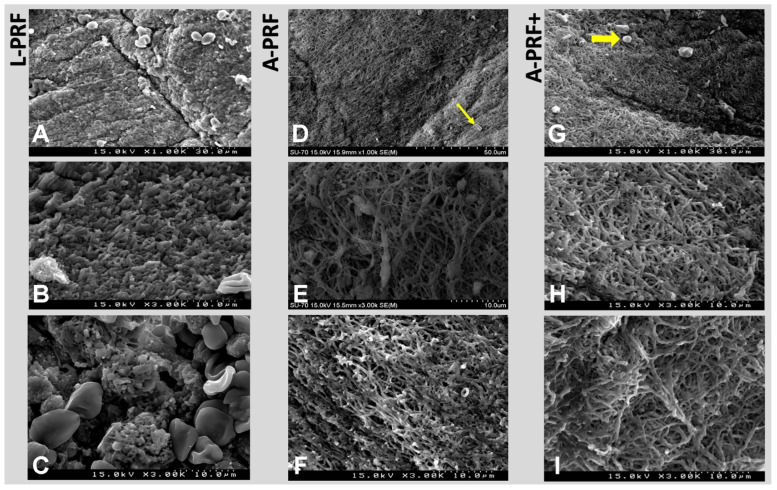
SEM images. (**A**–**C**) L-PRF: Matrix with a highly compact surface with thick fibers present within interfibrous areas as well as limited and reduced space for microvascularization (**A**,**B**). Destruction of red blood cells and leukocytes is visible (**C**). (**D**–**F**) A-PRF shows a dense matrix (**D**) composed of thin and elongated fibers that follow a preferential and orientated direction (**E**,**F**) in which the platelets are well-adhered. Porosity is also evident with a large diameter of the interfibrous spaces (**E**). Silica microparticles lying on the surface ((**D**), yellow arrow). (**G**–**I**) A-PRF+: The amount of fibers cross-linking is even more noticeable than the porosity as the presence of intact lymphocytes are adhered to the surface of the mesh (**G**).

**Figure 6 polymers-14-01392-f006:**
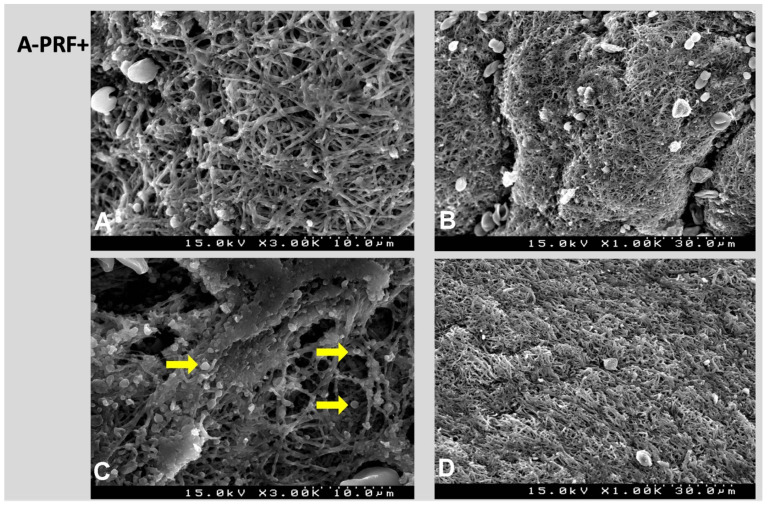
SEM images. A-PRF+ fibers and platelets together (**A**). It is possible to observe red cells and platelets adhering to the fibers ((**B**,**C**), yellow arrow); (**D**) shows a more fibrotic portion characterized by the lower-upper area.

## Data Availability

All data are available within this article.

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
