# Peer review of "Tensile Strength Essay Comparing Three Different Platelet-Rich Fibrin Membranes (L-PRF, A-PRF, and A-PRF+): A Mechanical and Structural In Vitro Evaluation"

_polymers, 2022, doi:10.3390/polym14071392_

Round 1

Reviewer 1 Report

Title: “Tensile strength essay comparing three different Platelet-Rich Fibrin membranes (L-PRF, A-PRF, and A-PRF+): mechanical and structural evaluation“

In this study the authors aimed at comparing the mechanical properties through tensile strength and analyze the structural organization among membranes produced by L-PRF (Leukocyte platelet-rich fibrin), A-PRF (Advanced Platelet-Rich Fibrin), and A-PRF+ (Advanced Platelet-Rich Fibrin plus) (original protocols), that vary in centrifugation speed and time. In particular, the authors submitted L-PRF (n=12), A-PRF (n=19), and A-PRF+ (n=13) membranes to the traction test, evaluating maximum and average tractions. Finally, the authors claim that this study concluded that A-PRF+ produced membranes with significant and higher maximum traction results, indicating a better viscoelastic strength when stretched by two opposing forces.

General comment: The topic of this study is interesting. Some issues should be reworked to improve the overall quality and impact of this work.

Some detailed comments:

Lines” 1 Graduated Dentist at the Integrated Master’s in Dental Medicine, Faculty of Dental Medicine at Universidade 7

Católica Portuguesa (Viseu, Portugal) 8

2 Professor of Periodontics; Center for Interdisciplinary Research in Health, Institute of Health Sciences, and 9

Faculty of Dental Medicine at Universidade Católica Portuguesa (Viseu, Portugal) (orcid: 0000-0001-6592-2290) 10

3 Associate Professor, Coordinator of TEMA - University of Aveiro (Portugal) (orcid: 0000-0002-3972-8432) 11

4 Department of Periodontics and Oral Medicine, University of Michigan, Ann Arbor, U.S.A. (orcid: 0000-0001- 12

5358-612X) 13

5 Department of Periodontics and Oral Medicine, University of Michigan, Ann Arbor, U.S.A. (orcid: 0000-0003- 14

3022-4390) 1”

*) Normally, only the affiliations (University of others) are shown. The current positions do not have matter…

Figure 4 and lines: “ Figure 4. A. Resistance and maximum traction, where A-PRF+ reach the higher strength value com- 224
paring with A-PRF and L~PRF tested; B. Average and standard deviation (SD) comparing all groups 225
analyzed; and C. Membrane resistance and median show the variability of results felt mainly on the 226
L-PRF group and more consistent in the A-PRF+. “

*) This key figure should be reworked. In particular, all subfigures are too small, please enlarge and magnify. The subfigure A is not clear since its labels are not meaningful (what is Log10 … is a logarithm of a strain or of a stretch or other, please clarify)

The viscoleastic properties (viscoelastic strength) of membranes are mentioned in the abstract. How does the“ viscoelastic strength” is defined here ? What is the specific test used to probe this properties here ? Please clarify.

Lines: “Tensile assay 279
In the tensile test, the overall evaluation tended towards discovering a quantitative 280
order in the resistance parameter, demonstrating the maximum tensile strength results for 281
A-PRF +, followed by A-PRF. Although Ravi and Santhanakrishnan39 found extremely
high values for A-PRF (362.565 ± 5.15 MPa), these values do not coincide with those pre- 283
sented in this study, with different protocol settings justifying. Samples were produced 284
with a tabletop centrifuge (Remi R4C) and cut according to a higher Length mold to per- 285
form the tensile test using a universal testing machine. 286
However, there was compliance with the scenario of a notable increase in the elastic 287
capacity when LSCC was implemented39. In the present work, A-PRF ranged the average 288
of 0.0020 MPa, and A-PRF+ obtained an average of 0.0022MPa, depending on a much 289
higher tensile force to suffer deformation or rupture, only by reduction of speed centrifu- 290
gation and time. Although both scored lesser than the obtained 0.2 ± 0.06 MPa resistance 291
of L-PRF prepared by Khorshidi et al.19, results carried out by the methodologies adopted 292
by Pascoal et al.37 recorded in A-PRF a value of 0.0752 MPa for maximum traction, higher 293
than what is shown in this study, but which still surpasses the reach of membranes pro- 294
duced by L-PRF. The motive for this result to be more elevated is the cut size of the mem- 295
brane, which was more significant than the standardized measures in our research. 296
The effect felt in the A-PRF+ group were approximately similar to A-PRF. However, 297
they cannot be entirely compared with previously reported studies since this study, as far 298
as it is shown in literature, is the first to elucidate the inherent mechanical properties of 299
this derivative using strictly standard protocol techniques, originally described12,40
Lam et al.21 refer to the action of platelets on the clot, detecting a positive impact 301
which, in addition to fibrin, are a source of reinforcement of the adhesive properties. This 302
fact suggests that the significant increase in elasticity and tensile strength found in the 303
low-RCF protocols performed may also result from the more dispersed platelet distribu- 304
tion along with the clot. In agreement with the previous reports1, which concluded that 305
the slight decrease in the centrifugation time preserves the cellular contents largely, pre- 306
senting a higher amount and an even more nearly homogeneously distribution of plate- 307
lets1,11. Therefore, this suggests that A-PRF+ could be preferable as a scaffold for tissue 308
repair when high tensile strength is an issue, such as in wound closure, distribution of 309
mechanical forces, and holding the suture.

*) The viscoleastic properties of membranes are mentioned in the abstract: where are they discussed in the work ? Where are they evaluated ? Please clarify…

Paragraphs : “Other considerations” and “Final considerations” and “

*) Perhaps it is better to merge these paragraphs into one

Author Response

Thank you for all your comments. Below, the responses.

  1. Normally, only the affiliations (University of others) are shown. The current positions do not have matter…

R: Thank you. It was adjusted.

  1. Figure 4 and lines. This key figure should be reworked. In particular, all subfigures are too small, please enlarge and magnify. The subfigure A is not clear since its labels are not meaningful (what is Log10 … is a logarithm of a strain or of a stretch or other, please clarify).

R: Thank you. Figure 4 was adjusted and corrected.

  1. The viscoleastic properties (viscoelastic strength) of membranes are mentioned in the abstract. How does the“ viscoelastic strength” is defined here? What is the specific test used to probe this properties here? Please clarify.

R: Viscoelasticity is the property of biomaterials that exhibit viscous and elastic characteristics when undergoing deformation. The tensile assay was the test performed, and it was done SEM images to verify deformation.

  1. Lines: Discussion - “Tensile assay”. The viscoleastic properties of membranes are mentioned in the abstract: where are they discussed in the work ? Where are they evaluated ? Please clarify…

R: Thank you for these questions. The viscoelastic property was evaluated through tensile test and SEM. We divided our discussion into two subtopics: Tensile assay and SEM analysis. The text involving this question is red.

  1. Paragraphs: “Other considerations” and “Final considerations”. Perhaps it is better to merge these paragraphs into one.

R: The subtopics were merged.

Reviewer 2 Report

The authors wrote an original article to compare the mechanical properties through tensile strength and analyze the structural organization among membranes produced by L-PRF (Leukocyte platelet-rich fibrin), A-PRF (Advanced Platelet-Rich Fibrin), and A-PRF+ (Advanced Platelet-Rich Fibrin plus).

The study is well written however for publication i have the following comments:

Title: please add the study design.

Abstract: Li 27 "All groups had significant statistical results." This sentence is unclear please revise.

Introduction: Please use the MDPI references style and order your references chronologically.

Material and Methods: Please add the name of the institution of the Ethics Committee and date of approval.

Please add sample size calculation

Discussion: Please discuss the future research step based on the study findings.

Please add the following points: 

Funding:

Conflicts of Interest:

Author Response

Thank you for all your comments. Below, are responses to the questions done.

  1. Title: please add the study design.

R: Thank you. “in vitro” was added to the title.

  1. Abstract: Li 27 "All groups had significant statistical results." This sentence is unclear please revise.

R: Thank you. The phrase was adjusted.

  1. Introduction: Please use the MDPI references style and order your references chronologically.

R: It was done as requested.

  1. Material and Methods: Please add the name of the institution of the Ethics Committee and date of approval.

R: The data were included.

  1. Please add sample size calculation

R: Sample size calculation was inserted in the M&M.

  1. Discussion: Please discuss the future research step based on the study findings.

R: It was included one paragraph within the discussion.

  1. Please add the following points: Funding and Conflicts of Interest.

R: These points were included after the conclusion.

Reviewer 3 Report

Paper titled (Tensile strength essay comparing three different Platelet-Rich Fibrin membranes (L-PRF, A-PRF, and A-PRF+): mechanical and structural evaluation) by Simões-Pedro et al. prepared and compared 3 Platelet rich fibrin membranes regarding their structure and also mechanical properties.

1- Give the origin of chemicals completely and consistently, code, company, town, state & country. Also version of software & instruments.

2- Statistical analysis: did the authors check the normality of distribution of the data using a suitable test such as Shapiro-Wilk test or K-S test before deciding to applyt test? This should be doen and written in methods clearly, as if NOT, authors should shift to non-parametric ANOVA

3- In methods, write the p value

4- In each figure legend where stat analysis was done, mention the p value and the stat test applied for these data.

Author Response

Reviewer 3: Thank you for all your comments.

1- Give the origin of chemicals completely and consistently, code, company, town, state & country. Also version of software & instruments.

R: It was provided. Please, let me know if there is any lack of information.

2- Statistical analysis: did the authors check the normality of distribution of the data using a suitable test such as Shapiro-Wilk test or K-S test before deciding to applyt test? This should be doen and written in methods clearly, as if NOT, authors should shift to non-parametric ANOVA

R: The data was inserted and adjusted.

3- In methods, write the p value

R: Thank you. The p value was included.

4- In each figure legend where stat analysis was done, mention the p value and the stat test applied for these data.

R: P-values were inserted in the respective figure.

Round 2

Reviewer 3 Report

Authors did not fully address the previous comments. Need to revise them again.  Overall I feel authors were not careful during the revision.

Also authors are requested to rewrite the last sentence in the methods

Also provide please the KS test results

Kriskal Wallis results and mention the post hoc test and show results I tabled as supp data but mention the F values or others in a table in the paper itself.

IAlso if data are not in normal dist, they should be presented as medians and quartile